A new minute ectosymbiotic harpacticoid copepod living on the sea cucumber Eupentacta fraudatrix in the East/Japan Sea

Yeom Jisu 1
Nikitin Mikhail A. 2
http://orcid.org/0000-0003-1255-0491 Ivanenko Viatcheslav N. 3
http://orcid.org/0000-0002-9873-1033 Lee Wonchoel 1 wlee@hanyang.ac.kr
1 Department of Life Science, Hanyang University , Seoul , South Korea
2 A.N. Belozersky Institute of Physico-chemical Biology, Lomonosov Moscow State University , Moscow , Russia
3 Department of Invertebrate Zoology, Biological Faculty, Lomonosov Moscow State University , Moscow , Russia
Costello Mark
Electronic publication date: 2018 Jun 14
Publication date: 2018
Volume: 6
Electronic Location ID: e4979
Received 2017 Dec 21; Accepted 2018 May 16
Copyright: © 2018 Yeom et al.
Copyright year: 2018
Copyright holder: Yeom et al.
License: This is an open access article distributed under the terms of the Creative Commons Attribution License, which permits unrestricted use, distribution, reproduction and adaptation in any medium and for any purpose provided that it is properly attributed. For attribution, the original author(s), title, publication source (PeerJ) and either DOI or URL of the article must be cited.
License URL: https://creativecommons.org/licenses/by/4.0/

Keywords: Copepoda, Laophontidae, Eupentacta fraudatrix, Ectosymbiosis, New genus, 18S rDNA

Funding: BK21 Plus Program (Eco-Bio Fusion Research Team) 22A20130012352 Ministry of Education (MOE, Korea) Russian Foundation for Basic Research #15-29-02601, #15-54-78061 and #18-04-01192 Russian Science Foundation # 14-50-00029 This study was supported by the BK21 Plus Program (Eco-Bio Fusion Research Team, 22A20130012352) funded by the Ministry of Education (MOE, Korea). Fieldwork, sorting of samples and molecular laboratory work were conducted with support from the Russian Foundation for Basic Research (#15-29-02601, #15-54-78061 and #18-04-01192, respectively). Molecular data analyses was supported by the Russian Science Foundation (# 14-50-00029). The funders had no role in study design, data collection and analysis, decision to publish, or preparation of the manuscript.

==============================
The ectosymbiotic copepods, Vostoklaophonte eupenta gen. & sp. nov. associated with the sea cucumber Eupentacta fraudatrix, was found in the subtidal zone of Peter the Great Bay, East/Japan Sea. The new genus, Vostoklaophonte, is similar to Microchelonia in the flattened body form, reduced mandible, maxillule and maxilla, but with well-developed prehensile maxilliped, and in the reduced segmentation and setation of legs 1–5. Most appendages of the new genus are more primitive than those of Microchelonia. The inclusion of the symbiotic genera Microchelonia and Vostoklaophonte gen. nov. in Laophontidae, as well as their close phylogenetic relationships, are supported by morphological observations and molecular data. This is the third record of laophontid harpacticoid copepods living in symbiosis with sea cucumbers recorded from the Korean and Californian coasts.

Introduction

Symbiotic harpacticoids that use holothurians as hosts are rarely reported compared to the orders Poecilostomatoida and Siphonostomatoida (Humes, 1980; Ho, 1982; Jangoux, 1990; Mahatma, Arbizu & Ivanenko, 2008; Avdeev, 2017). Among harpacticoids, only one species of Tisbidae Stebbing, 1910—Sacodiscus humesi Stock, 1960— and two species of Laophontidae T. Scott, 1905—Microchelonia californiensis (Ho & Perkins, 1977) and M. koreensis (Kim, 1991)—have been found associated with sea cucumbers (Huys, 2016).

Stock (1960) found S. humesi in washings of Holothuria tubulosa Gmelin, 1791 collected in the Bay of Banyuls. M. californiensis was found associated with the holothurian Apostichopus parvimensis (Clark, 1913) on the Californian coast. M. californiensis was originally described as Namakosiramia californiensis Ho & Perkins, 1977, and was designated by Ho & Perkins (1977) as the type of their newly established “siphonostome” cyclopoid family Namakosiramiidae. Ho (1986) concluded that Namakosiramiidae “should have been placed in the order Harpacticoida,” but its position within Harpacticoida remained unclear until Huys (1988) re-examined the type material of N. californiensis, removed the family from the Siphonostomatoida and placed it in the Harpacticoida, and relegated it to a junior subjective synonym of the family Laophontidae (see also Huys, 2009). The second species, M. koreensis (Kim, 1991), was found and described associated with the holothurian Apostichopus japonicus (Selenka, 1867) kept in the aquarium of a fish market in Gangneung at the Korean east coast (Kim, 1991).

The family Laophontidae consists of 325 valid species in 73 genera and two subfamilies (Walter & Boxshall, 2017) and includes diverse forms with cylindrical or dorsoventrally flattened bodies, and with reduced armature complement and segmentation of the legs (Gheerardyn et al., 2007).

During a survey of symbiotic copepods associated with invertebrates at Peter the Great Bay, East Sea (Japan Sea), a new harpacticoid copepod of the family Laophontidae associated with the sea cucumber Eupentacta fraudatrix (D’yakonov & Baranova in D’yakonov, Baranova & Savel’eva, 1958) is found and described herein.

Materials and Methods

The laophontid harpacticoid copepods Vostoklaophonte eupenta gen. & sp. nov. associated with the sea cucumber Eupentacta fraudatrix, and Microchelonia koreensis associated with the spiked sea cucumber A. japonicus, were collected on October 17 2013 at the subtidal zone of the “Vostok” research station at Peter the Great Bay of the East Sea (Japan Sea). 23 specimens of sea cucumbers (17 specimens of E. fraudatrix and five specimens of A. japonicus) were collected by hand. The sea cucumbers were placed in plastic bags and rinsed in situ with 10% ethanol to anesthetize and detach the copepods. The washings were sieved using a 60 μm sieve, and copepods were sorted with a pipette under an Olympus SZX 7 dissecting microscope. Copepods were fixed in 70% ethanol for morphological observation.

Copepods were dissected in lactic acid, and the dissected parts were mounted on slides using lactophenol as mounting medium. Preparations were sealed with transparent nail varnish. All drawings were prepared using a camera lucida on an Olympus BX51 differential interference contrast microscope.

Specimens for SEM micrographs were dehydrated through graded ethanol series, critical point dried, mounted on stubs and sputter-coated with platinum. The material was photographed using a Hitachi S-4700 scanning electron microscope at Eulji University, Seoul, Korea. All the specimens were deposited in the collection of the National Institute of Biological Resources, Korea (NIBR) and in the Zoological Museum of Lomonosov Moscow State University (ZMMU).

DNA was extracted from ethanol-preserved specimen using Diatom DNA Prep 100 kit (Isogene, Moscow, Russia). Nuclear 18S rDNA was amplified using Encyclo Plus PCR kit (Evrogen, Moscow, Russia) and universal primers Q5 and Q39 (Medlin et al., 1988). DNA amplification through PCR was as follows: 3 min at 95 °C, the 37 cycles of 94 °C for 20 s, annealing at 54 °C for 30 s, 72 °C for 1 min 30 s and final elongation at 72 °C for 5 min. PCR products were purified with preparative electrophoresis in 1% agarose gel. Bands of DNA of appropriate length were excised from gel and DNA was extracted using GelPrep spin-column kit (Cytokine). Extracted DNA was sequenced on ABI 3730 capillary sequencer from both ends.

Previously recorded sequences of nuclear 18S-rDNA from GenBank were aligned using the Muscle algorithm integrated in MEGA 6.0 (Edgar, 2004). Consequently, we generated an alignment of 1929 bp for 45 taxa (listed in Table 1) for 18S-rDNA. Models of nucleotide evolution were estimated using ModelGenerator (Keane et al., 2006). GTR+G+I model (General Time-Reversible with gamma distribution of rates across sites and proportion of invariant sites) was found optimal. Neighbor-joining trees were built in MEGA 6.0 (Tamura et al., 2013) and Bayesian phylogenetic trees were built in MrBayes 3.2.6 (Ronquist et al., 2012). Two Markov chain Monte Carlo (MCMC) chains were run in parallel and the analyses were stopped when average standard deviations of split frequencies between chains was below 0.01. 1,500,000 tree generations were produced Burn-in was set at 500,000 trees.

Table 1 GenBank numbers of sequences used in phylogenetic analyses in this study.

Name	Family	Accession no.	Reference	
Ameira scotti	Ameiridae	EU380303.1	Huys, Mackenzie-Dodds & Llewellyn-Hughes (2009)	
Sarsameira sp.	Ameiridae	EU380304.1	Huys, Mackenzie-Dodds & Llewellyn-Hughes (2009)	
Nitocra hibernica	Ameiridae	EU380305.1	Huys, Mackenzie-Dodds & Llewellyn-Hughes (2009)	
Cancrincola plumipes	Ameiridae	L81938.1	Spears & Abele (1998)	
Argestigens sp.	Argestidae	EU380306.1	Huys, Mackenzie-Dodds & Llewellyn-Hughes (2009)	
Eurycletodes laticauda	Argestidae	EU380310.1	Huys, Mackenzie-Dodds & Llewellyn-Hughes (2009)	
Bryocamptus pygmaeus	Canthocamptidae	AY627015.1	Huys et al. (2006)	
Attheyella crassa	Canthocamptidae	EU380307.1	Huys, Mackenzie-Dodds & Llewellyn-Hughes (2009)	
Mesochra rapiens	Canthocamptidae	EU380308.1	Huys, Mackenzie-Dodds & Llewellyn-Hughes (2009)	
Itunella muelleri	Canthocamptidae	EU380309.1	Huys, Mackenzie-Dodds & Llewellyn-Hughes (2009)	
Canuella perplexa	Canuellidae	EU370432.1	von Reumont et al. (2009)	
Dactylopusia sp.	Dactylopusiidae	EU380295.1	Huys, Mackenzie-Dodds & Llewellyn-Hughes (2009)	
Diarthrodes sp.	Dactylopusiidae	EU380296.1	Huys, Mackenzie-Dodds & Llewellyn-Hughes (2009)	
Sewellia tropica	Dactylopusiidae	EU380299.1	Huys, Mackenzie-Dodds & Llewellyn-Hughes (2009)	
Dactylopusia pauciarticulata	Dactylopusiidae	KR048735	S. Baek, 2015, unpublished data	
Bradya sp.	Ectinosomatidae	AY627016.1	Huys et al. (2006)	
Tigriopus japonicus	Harpacticidae	EU054307.1	Ki et al. (2009)	
Tigriopus fulvus	Harpacticidae	EU370430.1	von Reumont et al. (2009)	
Zaus caeruleus	Harpacticidae	EU380284.1	Huys, Mackenzie-Dodds & Llewellyn-Hughes (2009)	
Harpacticus sp.	Harpacticidae	EU380285.1	Huys, Mackenzie-Dodds & Llewellyn-Hughes (2009)	
Harpacticus nipponicus	Harpacticidae	KR048736	S. Baek, 2015, unpublished data	
Paralaophonte congenera	Laophontidae	KR048738	S. Baek, 2015, unpublished data	
Microchelonia koreensis	Laophontidae	MG012752	This study	
Vostoklaophonte eupenta	Laophontidae	MG012753	This study	
Laophontina sp.	Laophontidae	MF077713	Khodami et al. (2017)	
Pseudonychocamptus spinifer	Laophontidae	MF077714	Khodami et al. (2017)	
Lourinia armata	Louriniidae	KR048739	S. Baek, 2015, unpublished data	
Diosaccus sp.	Miraciidae	EU380290.1	Huys, Mackenzie-Dodds & Llewellyn-Hughes (2009)	
Stenhelia sp.	Miraciidae	EU380291.1	Huys, Mackenzie-Dodds & Llewellyn-Hughes (2009)	
Typhlamphiascus typhlops	Miraciidae	EU380292.1	Huys, Mackenzie-Dodds & Llewellyn-Hughes (2009)	
Paramphiascella fulvofasciata	Miraciidae	EU380293.1	Huys, Mackenzie-Dodds & Llewellyn-Hughes (2009)	
Miracia efferata	Miraciidae	EU380294.1	Huys, Mackenzie-Dodds & Llewellyn-Hughes (2009)	
Amonardia coreana	Miraciidae	KT030261	S. Baek, 2015, unpublished data	
Parastenhelia sp.	Parastenheliidae	EU380302.1	Huys, Mackenzie-Dodds & Llewellyn-Hughes (2009)	
Peltidium sp.	Peltidiidae	EU380288.1	Huys, Mackenzie-Dodds & Llewellyn-Hughes (2009)	
Alteuthellopsis sp.	Peltidiidae	EU380289.1	Huys, Mackenzie-Dodds & Llewellyn-Hughes (2009)	
Porcellidium ofunatense	Porcellidiidae	KR048741	S. Baek, 2015, unpublished data	
Euterpina acutifrons	Tachidiidae	GU969212.1	M. Wang, 2010, unpublished data	
Tegastes sp.	Tegastidae	EU380287.1	Huys, Mackenzie-Dodds & Llewellyn-Hughes (2009)	
Phyllothalestris sp.	Thalestridae	EU380298.1	Huys, Mackenzie-Dodds & Llewellyn-Hughes (2009)	
Paramenophia sp.	Thalestridae	EU380300.1	Huys, Mackenzie-Dodds & Llewellyn-Hughes (2009)	
Eudactylopus sp.	Thalestridae	EU380301.1	Huys, Mackenzie-Dodds & Llewellyn-Hughes (2009)	
Parathalestris parviseta	Thalestridae	KR048742	S. Baek, 2015, unpublished data	
Tisbe sp. 1	Tisbidae	FJ713566.1	Chullasorn et al. (2011)	
Tisbe sp. 2	Tisbidae	KR048743	S. Baek, 2015, unpublished data	

The descriptive terminology is adopted from Huys et al. (1996). Abbreviations used in the text are: A1, antennule; A2, antenna; ae, aesthetasc; exp, exopod; enp, endopod; P1–P6, first to sixth legs; exp (enp)-1(2, 3) denotes the proximal (middle, distal) segment of the exopod (endopod). Scale bars in figures are in μm.

The electronic version of this article in portable document format will represent a published work according to the International Commission on Zoological Nomenclature (ICZN), and hence the new names contained in the electronic version are effectively published under that Code from the electronic edition alone. This published work and the nomenclatural acts it contains have been registered in ZooBank, the online registration system for the ICZN. The ZooBank LSIDs (Life Science Identifiers) can be resolved and the associated information viewed through any standard web browser by appending the LSID to the prefix http://zoobank.org/. The LSID for this publication is: urn:lsid:zoobank.org:pub:4FDE5EAE-24A0-4320-A06C-1FD8F983A0BE. The online version of this work is archived and available from the following digital repositories: PeerJ, PubMed Central and CLOCKSS.

Systematics

Order Harpacticoida Sars, 1903

Family Laophontidae T. Scott, 1905

Subfamily Laophontinae T. Scott, 1905

Vostoklaophonte gen. nov.

urn:lsid:zoobank.org:act:1988C43D-50A0-4785-83CC-A3BB870A1972

Diagnosis. Laophontinae. Body dorsoventrally flattened; female genital field with two setae on P6 and small copulatory pore located in median depression; anal operculum well-developed. Sexual dimorphism in antennules, P3–P6, and genital segmentation. Rostrum large, rectangular and fused to cephalothorax; antennule six-segmented in female and seven-segmented subchirocer in male, aesthetascs present on segments 4 and 6 in female, on segments 5 and 7 in male; mandibular palp with four elements; coxal endite of the maxillule small with three elements; syncoxa of maxilliped with one element. P1 exopod two-segmented; P2 with three-segmented exopod and two-segmented endopod; P3 with three-segmented exopod and two-segmented endopod in the female, with two-segmented exopod and two-segmented endopod in the male; male P3 endopod without apophysis; P4 exopod one-segmented in female, two-segmented in male; P4 endopod one-segmented in both sexes; P5 exopod separated from baseoendopod in both sexes.

Etymology. The generic name refers to the type locality, the Vostok research station, and to the type genus of the family. Gender feminine.

Type species. V. eupenta gen. & sp. nov., by monotypy.

Vostoklaophonte eupenta sp. nov.

urn:lsid:zoobank.org:act:67348997-40CB-4C48-92F6-066BEBE90B67

Figs. 1–8

Figure 1 Vostoklaophonte eupenta gen. & sp. nov. (♀).

(A) Habitus, dorsal. Figure by Wonchoel Lee.

Figure 2 Vostoklaophonte eupenta gen. & sp. nov. (♀).

(A) Antennule, dorsal (setae omitted from segment 6). (B) Sixth segment of antennule. (C) Urosome, ventral (excluding somite bearing P5). (D) Anal somite and caudal rami, dorsal. Figure by Wonchoel Lee.

Figure 3 Vostoklaophonte eupenta gen. & sp. nov. (♀).

(A) Antenna. (B) Mandible. (C) Maxillule. (D) Maxilla. (E) Maxilliped. Figure by Wonchoel Lee.

Figure 4 Vostoklaophonte eupenta gen. & sp. nov. (♀).

(A) P1. (B) P2. (C) P3. (D) P4. (E) P5. Figure by Wonchoel Lee.

Figure 5 Vostoklaophonte eupenta gen. & sp. nov. (♂).

(A) Habitus, dorsal (B) Antennule (setae omitted from 5th & 7th segments). (C) 5th antennulary segment. (D) 7th antennulary segments. Figure by Wonchoel Lee.

Figure 6 Vostoklaophonte eupenta gen. & sp. nov. (♂).

(A) P2, anterior. (B) P3, anterior. (C) P4, anterior. (D) P5, anterior. (E) Urosome, ventral (excluding the first somite bearing P5). Figure by Wonchoel Lee.

Figure 7 Vostoklaophonte eupenta gen. & sp. nov. SEM photographs.

(A) P3 (♀, abnormal inner seta arrowed) (B) Genital area (♀, genital pore arrowed). (C) Caudal rami, ventral (♀, tube pore arrowed). (D) Antennule (♂). (E) Antenna (♂). (F) P2 & P3 (♂).

Figure 8 Phylogenetic tree of harpacticoids based on nuclear 18S ribosomal DNA data.

A 25% majority consensus of 1,500 trees generated using MBayes 3.2.6 (Ronquist et al., 2012) under the GTR+G+I model. Numbers at nodes represent Bayesian posterior probabilities. Members of the family Laophontidae showed in bold. Symbionts of holothurians are marked with asterisk (*).

Type locality. The subtidal zone at the Vostok research station (42°53′37.5′′N 132°44′00.9′′E), Peter the Great Bay, Russia, the East Sea (Japan Sea); 0.2–1 m depth; October 17, 2013.

Material examined. 1♀ holotype (NIBRIV0000812797) dissected on one slide. 15 paratypes as follows: 1♂ (NIBRIV0000812897) dissected on one slide, 1♀ (NIBRIV0000812898) dissected on seven slides, 1♀ (NIBRIV0000812899) dissected on ten slides, 2♀♀ and 1♂ (NIBRIV0000812900) preserved in 70% alcohol, 2♀♀ and 3 copepodites (ZMMU Me–1208) preserved in 70% alcohol. Four specimens (3♀♀ and 1♂) dried, mounted on stubs, and coated with gold for SEM (NIBRIV0000812901). All specimens are from the type locality.

Etymology. The specific name refers to the host of the new species, the holothurian Eupentacta fraudatrix.

DNA-barcode (18s rDNA). Sequence (1,929 base pairs) was submitted to GenBank (Genbank Accession number: MG012753.

Host. Sea cucumber, Eupentacta fraudatrix (Echinodermata: Holothuroidea: Dendrochirotida).

Description of female. Total body length of holotype measured from tip of rostrum to posterior margin of caudal rami 563 μm (paratypes, n = 3, mean = 583 μm). Maximum width of holotype 336 μm (paratypes, n = 3, mean = 331 μm) measured at posterior margin of cephalothorax. Body (Fig. 1A) dorsoventrally flattened with two egg sacs. Rostrum (Fig. 1A) well developed, large and rectangular with one pair of anterior sensilla. Prosome (Fig. 1A) four-segmented, comprising cephalothorax and three pedigerous somites; P1-bearing somite fused to cephalothorax. Length:width ratio of cephalothorax, 0.78, subrectangular, with denticles on dorsal surface and setules along lateral margin. Sensilla scattered on cephalothorax, rarely present on other somites. All pedigerous somites with denticles on dorsal surface, long setules along lateral and posterior margins (Fig. 1A). Urosome (Figs. 1A, 2C–2D and 7B) five-segmented, comprising P5-bearing somite, genital double-somite, two free abdominal somites, and anal somite. Genital double-somite wide, with row of long spinules arising from transverse surface ridge dorsally and laterally. Genital field (Fig. 2C) located ventrally near anterior margin of genital double-somite, with median genital pore (arrowed in Fig. 7B). P6 (Fig. 2C) forming single plate, with well-developed opercula closing off paired genital apertures, each leg represented by two naked setae. Anal somite 1.9 times as wide as long, with well-developed smooth anal operculum, sensilla associated to the anal operculum not visible (Figs. 1A and 2D).

Caudal rami (Figs. 2C–2D and 7C) parallel, widely separated; length:width ratio, 0.93 ventrally, 0.88 dorsally; dorsal surface smooth, with short row of subdistal inner spinules ventrally; with well-developed tube pore at outer distal corner (arrowed in Fig. 7C); with seven setae: seta I smallest; setae II and III well developed, naked; seta IV pinnate; seta V pinnate, well developed, longest; seta VI naked, arising at inner distal corner; seta VII naked, triarticulate at base.

Antennule (Figs. 2A and 2B) slender, six-segmented; segment 1 with rows of spinules along anterior lateral margin, and along near articulation with succeeding segment; segments 2 and 3 with one row of spinules along posterior margin; segment 4 with one bare seta plus one slender seta fused basally with aesthetasc, the latter two elements issuing from sub-cylindrical process; segment 6 with six setae with articulated bases, with apical acrothek consisting of aesthetasc and two naked setae. Armature formula: 1-[1], 2-[8], 3-[7], 4-[1 + (1+ae)], 5-[1], 6-[3 + 6 articulated setae + acrothek].

Antenna (Fig. 3A) comprising coxa, allobasis, and one-segmented endopod. Coxa small and naked. Allobasis with one pinnate abexopodal seta located midway inner margin. Exopod one-segmented with four pinnate setae. Endopod rectangular, slightly longer than allobasis, with proximal inner and subdistal outer spinules, armature consisting of three strong and one pinnate spines, and two bare and two pinnate setae.

Mandible (Fig. 3B) small, with elongated gnathobase armed with several sharp teeth. Mandibular palp two-segmented; proximal segment with one short inner and one long outer naked seta; distal segment with two distal naked setae.

Maxillule (Fig. 3C) Praecoxa thin and elongated, without ornamentation. Arthrite of praecoxa armed with several sharp, narrow and tooth-like elements. Coxal endite fused to basis, endopod and exopod, forming one reniform segment with one inner and two naked distal setae.

Maxilla (Fig. 3D) Syncoxa with subdistal row of outer spinules, with 1 slender element consisting of two fused spines. Allobasis produced into strong curved pinnate claw. Endopod incorporated into allobasis, represented by two naked setae.

Maxilliped (Fig. 3E) three-segmented. Syncoxa with 1 naked seta. Basis strong, ovoid, with row of spinules near outer distal end. Endopod drawn out into smooth, strong claw, the latter with one accessory naked seta and one tube pore proximally.

P1 (Fig. 4A) Coxa without ornamentation. Basis armed with 1 outer and 1 inner naked seta. Exopod two-segmented; exp-1 with one outer spine; exp-2 slightly longer than exp-1, with five setae/spines. Endopod large, two-segmented; enp-1 2.4 times as long as exopod, without ornamentation; enp-2 with one small accessory seta, one large robust claw and ornamented with inner and outer spinules.

P2 (Fig. 4B) Praecoxa triangular. Coxa without surface ornamentation. Basis with one outer pinnate seta, and row of spinules at base of outer basal seta and between rami. Exopod three-segmented, about two times as long as endopod; exp-1 with outer spinules and one stout outer spine; exp-2 with one stout outer spine, without additional ornamentation; exp-3 with four elements (two stout outer spines, one distal long, and one inner, short, naked seta). Endopod two-segmented; enp-1 larger than enp-2, with spinules as shown, without armature; enp-2 with some outer spinules and one distal bipinnate seta.

P3 (Figs. 4C and 7A) Coxa without ornamentation. Basis with spinules at based of outer seta. Exopod three-segmented, each segment with outer spinules as shown; exp-1 with one long, pinnate, outer spine; exp-2 with one stout, short, outer spine; exp-3 with two pinnate, outer spines, and two pinnate setae (one inner and one distal). Endopod two-segmented; first segment with outer spinules; second segment with outer spinules and two inner spinules; enp-1 with one inner pinnate seta; enp-2 with three pinnate setae (one inner and one distal seta, and one outer spine).

P4 (Fig. 4D) Coxa smooth, fused to somite. Basis with spinules at base of outer seta and between rami. Exopod 2.6 times as long as endopod. Exopod one-segmented, rectangular, twice as long as wide, with three distal and two outer pinnate setae; with dense rows of spinules as figured; with one secretory pore near median distal margin. Endopod one-segmented, cylindrical, with one pinnate distal seta, and one row of spinules along outer margin.

Table 2 Armature formulae for P2–P4.

	Exopod	Endopod	
P2	0.0.022	0.010	
P3	0.0.022(0.113 in ♂)	1.111(0.020 in ♂)	
P4	032(0.121 in ♂)	010	

Armature formula as in Table 2.

P5 (Fig. 4E) Baseoendopod and exopod ornamented with spinules as shown. Baseoendopod with outer basal, naked seta. Endopodal lobe small, with two pinnate setae. Exopod rectangular, with five pinnate setae.

Description of male. Body (Fig. 5A) dorsoventrally flattened; total body length measured from tip of rostrum to posterior margin of caudal rami ranging from 366 μm to 400 μm (n = 2). Maximum width measured at posterior margin of cephalothorax ranging from 208 μm to 232 μm (n = 2). General body shape and ornamentation as in female except for lack of sensilla on cephalothorax. Sexual dimorphisms expressed in A1, P2, P3, P4, P5, P6 and genital field. One spermatophore present as in Fig. 5A.

Antennule (Figs. 5B–5D and 7D) seven-segmented, robust, subchirocer; segment-1 with row of inner spinules; segment four smallest, and incomplete sclerite with only 1 small seta; segment five swollen, largest, with two modified spines (1 thick, 1 short and trifid); segments 5 and 7 with aesthetasc. Armature formula; 1-[1], 2-[9], 3-[6], 4-[1], 5-[9 + 2 modified + (1+ae)], 6-[1], 7-[7 + acrothek]. Apical acrothek consisting of aesthetasc and two naked setae.

Antenna (Fig. 7E), mandible, maxillule, maxilla and maxilliped (not shown) as in female.

P1 (not shown) as in female.

P2 (Figs. 6A and 7F) Coxa with spinules close to joint with basis. Basis as in female, except for additional pore and lack of spinules between rami. Exopod as in female except for 1 spinular row only on exp-1, and for some spinules on exp-2 and -3. Endopod as in female, except for lack of spinules on enp-1.

P3 (Figs. 6B and 7F) Basis with some spinules at base of outer seta. Exopod two-segmented; outer spines more robust and shorter than in female; exp-1 with outer spinules, with one stout outer, pinnate spine; exp-2 with one inner, one distal, and three outer pinnate elements. Endopod two-segmented, without apophysis; enp-1 ornamented with one row of outer spinules distally, without armature; enp-2 with some inner spinules midway inner margin, with two distal pinnate setae.

P4 (Fig. 6C) Coxa without ornamentation. Basis with some spinules at base of outer seta. Exopod two-segmented; exp-1 with one pinnate outer spine and one row of outer spinules; exp-2 with one inner and two distal elements, with one outer pinnate spine, and with outer and inner spinules. Endopod one-segmented, trapezoid with one pinnate distal seta.

P5 (Fig. 6D) fused to somite. Baseoendopod with one pinnate outer basal seta, and endopodal lobe represented by one pinnate seta. Exopod small, rectangular, with one outer naked and three distal pinnate setae.

P6 (Fig. 6E) asymmetrical, represented on both sides by small plate (only left one functional); outer distal corner with one seta issuing from long setophore ornamented with some spinules.

Variability

A one-segmented mandibular palp with 4 elements (not shown) was observed in a female paratype specimen (destroyed during the observation). An abnormal short inner seta was observed in P3 exp-3 of paratype NIBRIV0000812901 (as arrowed in Fig. 7A).

Phylogenetic position

It is difficult to suggest a phylogenetic position of the new genus based on morphological characters due to their extreme reductions of mouthparts, and unusual sexual dimorphisms in the legs. However, a sister group relationship between Vostoklaophonte and Microchelonia can be hypothesized based on the 18S rDNA gene.

The phylogenetic trees based on the nuclear 18S rDNA gene (Fig. 8) shows the five genera of the family Laophontidae (Paralaophonte, Pseudonychocamptus, Laophontina, Microchelonia, Vostoklaophonte) are grouped together with very high support (98% bootstrap support in NJ tree and 99% Bayesian posterior probability in Bayesian tree). The high support (100%) observed for Vostoklaophonte gen. nov. and Microchelonia suggests a close relationship between these two genera.

Discussion

The new genus, Vostoklaophonte, is attributed here to the family Laophontidae T. Scott, 1905 as diagnosed by (Boxshall & Halsey, 2004), based on the presence of the following characters: (1) the six-segmented female antennule, and seven-segmented subchirocer in the male, (2) one abexopodal seta on the antennary endopod, and four elements on the one-segmented antennary exopod, (3) one seta only on the syncoxa of maxilliped, (4) P1 with large prehensile endopod and small exopod, (5) sexual dimorphism in antennules, genital segmentation and P5 and P6. Furthermore the new genus fits the diagnosis of the subfamily Laophontinae T. Scott, 1905 given by Huys & Lee (2000).

Vostoklaophonte eupenta displays the following unique combination of characters: (1) body dorso-ventrally flattened, (2) mouth parts highly reduced except for the well-developed maxillipeds, and (3) sexually dimorphic setation and segmentation of P2–P4. In addition, V. eupenta has synapomorphies including two segments distal to geniculation in the male antennule, maxillipedal syncoxa with one seta, the first endopodal segment of P1 without inner seta, the second endopodal segment of P2 without outer spine, and the endopod P3 of male without proximal inner seta in the female endopod as a member of Laophontinae.

Brady (1918) established the new genus Microchelonia for M. glacialis Brady, 1918 found in washing of Laminaria from Macquarie Island in the southwest Pacific Ocean. Boxshall & Halsey (2004) listed the genus Microchelonia in their list of “generic names—not in current use” without clear reason. Huys (2009) suggested that the genus Microchelonia belongs to the family Laophontidae and considered this genus a senior subjective synonym of Namakosiramia. Huys (2009) also wrote that Namakosiramia is the junior objective synonym of Microchelonia. Later on, Huys (2016) suggested that the family Namakosiramiidae is a junior synonym of the family Laophontidae. However, it was Huys (1988) who proposed that Namakosiramia should be placed in the Laophntidae: Laophontinae, and that Namakosiramiidae should be regarded as a synonym of Laophontidae. In his key to the species of Microchelonia, Huys (2016) included only two species, M. californiensis and M. koreensis because “the description of M. glacialis is grossly inadequate and its host is as yet unknown.”

The new genus is similar to the genera Peltidiphonte Gheerardyn & Fiers, 2006 and Microchelonia Brady, 1918 in having dorso-ventrally compressed body form, and the genera Afrolaophonte Chappuis, 1960 and Aequinoctiella Cottarelli, Bruno & Berera, 2008 in having reduced postmaxillipedal legs.

Vostoklaophonte seems to be closely related to Microchelonia by the flattened body form, the reduced mandible, maxillule, and maxilla, but well-developed maxilliped, and by the reduced segmentation and setation of P1–P4. The most appendages of the new genus seem to be more primitive than those of Microchelonia. For example, (1) the female antennule of the new genus is six-segmented, but four-segemented in Microchelonia, (2) the male antennule is seven-segmented in Vostoklaophonte, but six-segmented in Microchelonia, (3) the mandible, maxillule, and maxilla of the new genus possess more setae than those of Microchelonia, (4) the mandibular palp of Vostoklaophonte possesses four elements (see Fig. 3B), instead of with two as in Microchelonia (Ho & Perkins, 1977: 370; Huys, 1988: 1519, and Kim, 1991: 431, and Fig. 2C, present study), (5) the maxillule of Microchelonia is strongly reduced and is represented by an elongated arthrite bearing four speines (Ho & Perkins, 1977: 370, Huys, 1988: 1519, and Kim, 1991: 431, Fig. 2D), but maxillule with one-segmented coxa bearing three elements in Vostoklaophonte (see Fig. 3C), (6) the maxillary syncoxa possesses one endite in Vostoklaophonte but maxillary syncoxa without endites in Microchelonia (Huys, 1988: 1519). On the contrary, some appendages of the new genus seem to be more derived than in Microchelonia. For example, (1) the antennary exopod has four setal elements in both genera, but the distal spine on the endopods is reduced in Vostoklaophonte, and more developed in Microchelonia, (2) the maxilla is similar in both genera, except for the endopod represented by two setae in Vostoklaophonte, but represented by three setae in Microchelonia koreensis (Fig. 2E in Kim, 1991, p. 431, and Fig. 3D in this study), and (3) the maxilliped is well developed and stout in both genera, but the maxilliped of Microchelonia possesses more dense spinular patches than in the new genus (compare M. californiensis in Ho & Perkins (1977: 369, Fig. 7) and in Huys (1988: 1523, Fig. 3F), M. koreensis in Kim (1991: 431, Fig. 2F), and Vostoklaophonte (Fig. 3E) in present study).

Some other differences between Vostoklaophonte and Microchelonia were detected. The exopod of P1 is one-segmented with five elements in Microchelonia, but two-segmented with a total of six elements in Vostoklaophonte (compare Ho & Perkins (1977: 369, Fig. 8), Huys (1988: 1524, Fig. 4A) and Kim (1991, Fig. 2G), and Fig. 4A in the present study). The endopod of P1 is two-segmented and possesses a distal claw in the second segment in both genera, but spinules are present on the coxa and basis of Microchelonia only (compare Ho & Perkins (1977: 369, Fig. 8), Huys (1988: 1524, Fig. 4A) and Kim (1991, Fig. 2G), and Fig. 4A in the present study). Contrary to what has been observed in the new genus and species herein proposed, Microchelonia displays extreme reductions in P2–P4. Also, sexual dimorphism of Microchelonia is expressed in the relative length of the setae on P2–P4 (Kim, 1991, Figs. 2H–2J and 3C–3D), and in armature complement of P5 and P6 (Kim, 1991, Figs. 2K–2L and 3F–3G), but sexual dimorphism in Vostoklaophonte is expressed in P3 and P4 (e.g., the exopod of P3 is three-segmented in the female, but two-segmented in the male; the endopod of P3 in both sexes is two-segmented, but the male P3 endopod possesses a reduced number of setae on both segments, and based on the position of its setae, the two-segmented P3 exopod of male is most probably the result of the fusion of P3 exp-3 and exp-2 of the female; the exopod of P4 is one-segmented in the female, but two-segmented in the male. The exopod of P4 possesses five setae in both sexes, but the homologous setae are difficult to define), and no significant dimorphism was observed in P1 and P2. The exopod of P5 is clearly separated from the baseoendopod and possesses the five setae in the female, and four in the male. P6 is armed with two setae in the female and one seta in the male, similar to the condition observed for Microchelonia, and also typical for other family members.

Besides Microchelonia and Vostoklaophonte the flatten body form is also present in Peltidiphonte (Gheerardyn et al., 2006a). However, Peltidiphonte possesses well-developed mouthparts and swimming legs. Peltidiphonte also displays no sexual dimorphism in mouthparts and P1–P4 and possesses a spinous process on the second antennular segment. This suggests that Peltidiphonte is not closely related to the new genus, and the flattened body shape in these two genera must be the result of convergence.

Paralaophonte harpagone Gheerardyn, Fiers, Vincx & De Troch, 2006 has stout maxillipeds. The other shared features with Vostoklaophonte and Microchelonia include the rectangular rostrum, the number of segments of antennule in both sexes, the number of setae on the antennary exopod, the mandibular palp with only four elements, the two-segmented endopod of P1. The species has more primitive segmentation of P2–P4 than that of the two highly derived symbiotic genera. Since there are too many reductions in mouthparts and legs in Vostoklaophonte and Microchelonia, it is premature to claim that they are close to Paralaophonte lineage (Gheerardyn et al., 2006b).

The reduction of segmentation in P1–P4 found in several interstitial laophontids is different from that of Vostoklaophonte and Microchelonia. Aequinoctiella has one segmented exopod in P1–P4, no endopod in P2–P4, and P1 with two-segmented endopod (Cottarelli, Bruno & Berera, 2008).

Some morphological features shared by Vostoklaophonte and Microchelonia and the results 18s rDNA sequences (Fig. 8) suggest a close relationship between these two genera. However it is premature to claim a sister-group relationship or presence of a monophyletic lineage of symbiotic laophontids due to the lack of molecular data for most genera of the subfamily Laophontinae and for a number of undescribed symbiotic laophontids present in our collection.

Supplemental Information

Supplemental Information 1 Supplemental Information.

Click here for additional data file.

Authors also thanks to Raehyuk Jeong for reading and correcting the manuscript in English, Prof. Andrei Adrianov and Dr. Anton Chichvarkhin (A.V. Zhirmunsky Institute of Marine Biology) hosted MN and VI during the field trip. We express our sincere gratitude to editor Prof. Mark J. Costello, reviewer Prof. Samuel Gomez, and two anonymous reviewers for their constructive comments and advice on the earlier version of the manuscript.

Additional Information and Declarations

Competing Interests

Author Contributions

DNA Deposition

Data Availability

New Species Registration

The authors declare that they have no competing interests.

Jisu Yeom performed the experiments, analyzed the data, contributed reagents/materials/analysis tools, prepared figures and/or tables, authored or reviewed drafts of the paper, approved the final draft, processes for SEM photos.

Mikhail A. Nikitin performed the experiments, analyzed the data, contributed reagents/materials/analysis tools, authored or reviewed drafts of the paper, approved the final draft, analysis with 18s rDNA.

Viatcheslav N. Ivanenko conceived and designed the experiments, analyzed the data, contributed reagents/materials/analysis tools, authored or reviewed drafts of the paper, approved the final draft.

Wonchoel Lee conceived and designed the experiments, performed the experiments, analyzed the data, contributed reagents/materials/analysis tools, prepared figures and/or tables, authored or reviewed drafts of the paper, approved the final draft.

The following information was supplied regarding the deposition of DNA sequences:

The 18s rDNA sequences analyzed are accessible via Gen Bank accession number MG012753 and in the Supplemental Information.

The following information was supplied regarding data availability:

Holotype and parpatypes are deposited in the NIBR (National Institute of Biological Resources, Korea) Holotype 1♀ (NIBRIV0000812797) dissected on one slide. Paratypes 1♂ (NIBRIV0000812897) dissected on one slide, 1♀ (NIBRIV0000812898) on seven slides, 1♀ (NIBRIV0000812899) on 10 slides, 2♀♀, 1♂ (NIBRIV0000812900) in 70% alcohol, 2♀♀, 3 copepodites (Me-1208) in 70% alcohol. Four specimens (3♀♀, 1♂) dried, mounted on stubs, and coated with gold for SEM (NIBRIV0000812901). A part of paratypes (ZMMU Me-1208) are deposited in the Zoological Museum of Lomonosov Moscow State University.

The following information was supplied regarding the registration of a newly described species:

Publication LSID: urn:lsid:zoobank.org:pub:4FDE5EAE-24A0-4320-A06C-1FD8F983A0BE;

Vostoklaophonte gen. nov.

urn:lsid:zoobank.org:act:1988C43D-50A0-4785-83CC-A3BB870A1972;

Vostoklaophonte eupenta gen. & sp. nov.

urn:lsid:zoobank.org:act:67348997-40CB-4C48-92F6-066BEBE90B67.

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
