# Peer review of "A new minute ectosymbiotic harpacticoid copepod living on the sea cucumber Eupentacta fraudatrix in the East/Japan Sea"

_PeerJ, doi:10.7717/peerj.4979_

## Round 0.1 · original submission · Major Revisions

The referees all agree this is important work to be published. However, it is dissapointing that they found so many mistakes of an editorial nature. Also it is necessary to state which are type specimens. I'd ask you are painstaking in attending to the corrections asked, but also in checking for others the referees may have overlooked or may be created in doing the revisions. Because overall so many changes are recommended I am classifying this as Major Revision which means it will need to go to referees a second time. So please make them pleased with the next version.

·

Basic reporting

This is a nice and interesting contribution to the diversity of harpacticoid copepods of the family Laophontidae from the littoral zone of the East Japan Sea. The manuscript is relatively well-written, but in my opinion, the authors need to revise the general style.

I’m not an English native speaker but took the liberty to suggest some corrections to improve the general style.

It would be good to let an English native speaker read the manuscript to further improve the general style of the manuscript.

Experimental design

The manuscript fits the subjects of the journal, the question addressed by the authors is well-defined and relevant.

The authors describe a new genus and species of Laophontidae of the subfamily Laophontinae associated to a holothurian. This is interesting since only few symbiotic Laophontinae copepods have been reported associated to holothurians, filling an important gap to better understand the ecological radiation of laophontids. The investigation performed by the authors were conducted rigorously and meet the standards in harpacticoid taxonomy. The methods were described with sufficient detail.

The authors included the most relevant literature. However, in the list of references I detected that one of them is probably wrong (Huys R. 2009. On the junior subjective synonyms of Coullia Hamond, 1973 (Copepoda, Harpacticoida, Laophontidae): an update and key to species and related genera. Zookeys 5:33-40). I think the reference here is “Huys, R. 2009. Unresolved cases of type fixation, synonymy and homonymy in harpacticoid copepod nomenclature (Crustacea: Copepoda). Zootaxa 2183: 1-99”. I also suggested to include some other references that might help to improve the general content of the manuscript.

The figures included in the manuscript are very informative and of excellent quality. I suggest including a figure (map) showing the type locality, as well as the places where the species of Microchelonia have been found. I think this is important since the authors claim a sister group relationship between the new genus, Vostoklaophonte, and Microchelonia.

The general structure of the manuscript meets the requirements of the International Code of Zoological Nomenclature, the type material has been deposited in a well-known collection and is available for future inspection. However, I noted that some material was deposited in a collection whose acronym seems to be “Me”. I’m not sure if this acronym corresponds to the Zoological Museum of Lomonosov Moscow State University. This should be clearly stated in “Materials and Methods”.

Validity of the findings

The data are robust, and the results and conclusions are, in my opinion, sound. However, I still do not understand why the authors included the 18S-rDNA data for all the harpacticoid species available from GenBank. The information in Table 1 and in Figure 8 were not discussed in detail and, in my opinion, are not informative or relevant to the general content of the manuscript. Some comments on this and on other subjects were also included in the revised word document of the manuscript.

Additional comments

I have no general comments for the author(s) other that those in the revised word version of the manuscript.

Reviewer 2 ·

Basic reporting

A new genus and species of harpacticoid copepods is well described and illustrated. However, Discussion and others must be greatly revised. If SEM materials are not designated as paratypes, descriptions based on it should not be included in the descriptive parts. These are separately mentioned. Generally, a description of a new taxon must be based on type series (ICZN). Indicate what types were used for the illustrations. A molecular analysis should be reconsidered. In the descriptive parts, articles can be deleted for simple telegraphic style.

Experimental design

Terminology should be reconsidered. Since “The East Sea” is not internationally and officially recognized at the moment, it should not be used here. This is a scientific paper, but not a political one.

Validity of the findings

The definition of the new genus should be more cleared in comparison of other genera belonging to the subfamily Laophontinae. The new genus is compared with Microchelonia and others, but still problematic (see General comments).

Additional comments

See above also (Basic reporting, Experimental design, Validity of the findings).
Specific comments
1)Abstract. Laophontinae should be used here rather than Laophontidae (see Discussion).
You focused the subfamily rather than the family.
2) L41, 47. ..on the ...coast.
3) L49-52. This paragraph abruptly appears. Reconsider the context.
4) L65. Why did you use 10 % ethanol? If you followed previous reports, cite them.
5) L68, 69. This paragraph should be moved to somewhere in L80 to 98.
6) L77. Delete “All the...of”.
7) L104, 107. What article in ICZN?
8) Diagnosis. Some are not diagnostic to the new genus, but to Copepoda!
9) L135. You may use “by monotypy”.
10) L153. Citation is OK?
11) L156, 240, 242. Indicate what types were measured. Those of the holotypes should be mentioned separately.
12) L164. What type for 0.78?
13) L172. What do you mean by “Each of P6”?
14) L174. Delete “, but...unclear”.
15) L177. What type(s) for 0.93 and 0.88?
16) L188-192. Reconsider the description of the antenna. Exopod seems to be 2-segmented.
17) L195. Can you identify an outer seta as a remnant of the exopod?
18) L198, 199. The homology of the maxilla is OK? The conditions are highly reduced, your understanding of the homologues must be careful.
19) L200. Space is needed in front of Maxilla (new paragraph).
20) L204. ...drawn out into...
21) L218. Not plumose.
22) L251. Not only the antenna but also other mouthpart appendages should be mentioned in comparison with those of the female.
23) Phylogenetic position. Using numerous taxa, you said only one thing. Reconsider the analytic results and discussion.
24) Discussion. This part must be greatly changed for better context. The following steps may be better: (1) why can the present new genus belong to the subfamily?; (2) listing symapomorphies, closely related genera are compared to the new genus, and then differences among them must be stressed for the establishment of it; (3) some kinds of morphoclines throughout the symbiotic evolution of the subfamily can be discussed. L295-301 and L345-352 are strange considering the context. L326, 327. The new genus and Microchelonia are not properly compared here. L330. P3 and P4. L342. Don’t use “swimming”. L358. Microchelonia glacialis was found from Laminaria (L296).
25) Table 1.There are careless mistakes. Sp. is not italic; period of et al.
26) Fig. 1. This is an ovigerous female. So the egg-sac and eggs should be mentioned in the text.
27) Fig. 4D. This may be abnormal seeing that of the male. Check other types.

Reviewer 3 ·

Basic reporting

General things for English and sentences throughout the MS:
Some places are not available. Comma and space are not used proper. Hyphen and n-dash are also not utilized properly. Moreover, the unity of writing is not taken at all in the Descriptive section (e.g., three-segmented or 3-segmented). There are also many careless mistakes.


Line drawings:
Several points should be redrawn.

Experimental design

The main frame of this study is simple, and thus it is no problem.

Validity of the findings

The most important thing of the MS is to prove the validity of the new genus. However, this MS is inadequate for its explanation. Together with the Basic reporting, the MS requires to be polished before publishing.

Additional comments

The manuscript is including a description of new genus of symbiotic Harpacticoid. Descriptive part was prepared to explain the adequacy of the new genus, and subsequent phylogenetic analysis helped the result. However, there are several problems which have to be resolve before publishing.

Also, as I mentioned in the Basic reporting, there are many careless mistakes which I marked on the PDF file.

Annotated reviews are not available for download in order to protect the identity of reviewers who chose to remain anonymous.

---

## Round 0.2 · Minor Revisions

The MS is improved thank you. However, the referee (only one replied to our invitation) mentions some possible improvements which I think you can clarify to make it clearer for readers.

·

Basic reporting

No comment

Experimental design

No comment

Validity of the findings

No comment

Additional comments

This is the second review of the manuscript "A new ectosymbiotic harpacticoid copepod living on the sea cucumber Eupenctata fraudatrix in the East/Japan Sea". The authors did a good job with the "Introduction", "Materials and Methods" and "Systematics". I am not an English native speaker but took the liberty to suggest some changes in order to improve the manuscript. Also, I have only some minor comments. For example, I noted that the authors included in the abstract a new record of Microchelonia koreensis, but nothing is said about this in any other part of the manuscript. I still don't understand why the authors included all the harpactioid species in Table 1 and Figure 8. None of these species were discussed in the manuscript. Also, I suggest to check carefully the journal's format. For example, I'm not sure about the use of "1", "2", "3", "4", etc., instead of "one", "two", "three", "four", etc., "1-segmented" instead of "one-segmented" (the authors used both formats), etc.; also I suggest the authors to check that the authority of genera and species are given in full only the first time they are mentioned in the text, and to check the proper use of hypehns, n-dashes, and m-dashes (I assume that the use m-dashes should be kept at a minumum).
On the other hand, I think that the Discussion needs some extra work to improve the general content and structure of the manuscript. I took the liberty to suggest some changes. I am particularly interested in the possibility of (1) defining the new genus and species based on autapomorphies rather than on the combination of a set of characters, and (2) defining a monophyletic group of symbiotic genera (Vostoklaophonte and Microchelonia) based on synapomorphies and supported also on molecular evidence given by the authors. I think this could add some extra value to the manuscript.

---

## Round 0.3 · accepted · Accept

Thank you for attending to the minor revisions suggested by the referees. I think the paper reads well now and is an important contribution to the discovery of new species.